# CRISPR loss of function screening to identify genes involved in human primordial germ cell-like cell development

Young Sun Hwang[1], Yasunari Seita[1], M. Andrés Blanco[1,2]*, Kotaro Sasaki[1,2,3]*

**1** Department of Biomedical Sciences, University of Pennsylvania, School of Veterinary Medicine, Philadelphia, Pennsylvania, United States of America, **2** Institute for Regenerative Medicine, University of Pennsylvania, Philadelphia, Pennsylvania, United States of America, **3** Department of Pathology and Laboratory Medicine, University of Pennsylvania, Perelman School of Medicine, Philadelphia, Pennsylvania, United States of America

* ablanco@vet.upenn.edu (MAB); ksasaki@upenn.edu (KS)

**Data Availability Statement:** The datasets generated in this study are available at NCBI GEO under the following accession numbers: GSE202996.

## Abstract

Despite our increasing knowledge of molecular mechanisms guiding various aspects of human reproduction, those underlying human primordial germ cell (PGC) development remain largely unknown. Here, we conducted custom CRISPR screening in an *in vitro* system of human PGC-like cells (hPGCLCs) to identify genes required for acquisition and maintenance of PGC fate. Amongst our candidates, we identified *TCL1A*, an AKT coactivator. Functional assessment in our *in vitro* hPGCLCs system revealed that TCL1A played a critical role in later stages of hPGCLC development. Moreover, we found that TCL1A loss reduced AKT-mTOR signaling, downregulated expression of genes related to translational control, and subsequently led to a reduction in global protein synthesis and proliferation. Together, our study highlights the utility of CRISPR screening for human in vitro-derived germ cells and identifies novel translational regulators critical for hPGCLC development.

## Author summary

Proper development of the germline lineage into sperm and oocytes is required for propagation of inherited genetic information across generations. Accordingly, its abnormal development in humans results in a variety of medical conditions including infertility and congenital diseases. Despite its importance to human health, a comprehensive understanding of the mechanisms regulating human germ cell development is limited by both technical challenges and ethical concerns. The germline is established early in development in primordial germ cells (PGCs), the common precursor for both spermatozoa and oocytes. Notably, recent generation of PGC-like cells (hPGCLCs) derived from pluripotent stem cells allows for an unprecedented investigation of molecular mechanisms driving early germline development in humans. Using CRISPR loss-of-function screening, we now provide functional genomics resources critical for understanding hPGCLCs development. Using this screening technique, we have identified *TCL1A* as a novel gene that is critical for maintaining hPGCLCs. Functional validation analyses revealed that TCL1A

**Funding:** This work was supported in part by the Silicon Valley Community Foundation (2019-197906), Good Ventures Foundation (10080664) and the 2018 Health Research formula fund from the Commonwealth of Pennsylvania (#67-80) to K. S. The funders had no role in study design, data collection and analysis, decision to publish, or preparation of the manuscript.

**Competing interests:** The authors declare that they have no conflict of interest.

promotes global protein synthesis and cell proliferation through AKT-mTOR signaling pathways in hPGCLCs. This study highlights the utility of CRISPR screening in deciphering the genetic basis of human germ cell development.

## Introduction

Germ cells, including spermatozoa and oocytes, are critical for transmission of genetic and epigenetic information to the next generation. Primordial germ cells (PGCs), which are the early-stage precursors of spermatozoa and oocytes, are specified during early embryonic development through dynamic processes initiated by inductive signals from neighboring tissues. In humans, abnormal germline development leads to a variety of medical conditions, including infertility and congenital diseases. Thus, a comprehensive understanding of the mechanisms regulating germ cell development has critical significance for human health.

Mammalian germ cell development has been extensively studied in mice [1]. However, early embryogenesis differs between mouse and humans [2], raising concerns as to the translational relevance of such findings to human germ cell development. Indeed, analysis of the specification of PGCs in cynomolgus monkeys (*Macaca fascicularis*), which exhibit a planar structure of peri-implantation embryo, supports the idea that primate PGC specification differs substantially from that in mice with respect to both their origin and developmental trajectories [3]. However, analyses of mechanisms underlying germ cell development in humans are hampered by the technical and ethical difficulties in accessing scarce developing germ cells that are only transiently present in human fetal material. Notably, recent advances in an *in vitro* model of human PGCs, termed PGC-like cells (hPGCLCs), derived from human ESCs or iPSCs [4,5], have provided a foundation for investigating underlying mechanisms that is not restricted by the accessibility and ethical difficulties of research in intact human embryos.

CRISPR loss-of-function genetic screening has been used to discover novel molecular regulators of a wide range of cellular phenotypes in a variety of biological contexts [6,7], including PGCLC specification from mouse ESCs [8]. However, to date no functional screens for regulators of human germline development and specification have been reported. Here, we carried out a customized CRISPR screen in an *in vitro* system of hPGCLCs derived from a highly quantifiable reporter human inducible pluripotent stem cell (hiPSC) line. In addition, we further elucidated the role of a novel candidate, the AKT coactivator TCL1A, in promoting hPGCLC development.

## Results

### Generation of inducible Cas9 expressing hiPSCs (17C-2 iCas9)

To enable CRISPR loss-of-function genetic screening for regulators of hPGCLC specification and development, we first established an inducible Cas9 (iCas9) hiPSC cell line in which gene editing would occur following hPGCLC induction, but not in the hiPSC state. For this, we utilized a *piggyBac* vector (PB-iCas9-Neo), which encodes both a FLAG-tagged Cas9 and reverse tetracycline-controlled transactivator (rtTA) to confer Doxycycline (Dox)-responsive Cas9 expression, and a Transposase vector for vector integration (**Fig 1A, Materials and methods**). These vectors were introduced into *TFAP2C-2A-EGFP* and *DDX4/hVH-2A-tdTomato* (AGVT) hiPSCs (585B1 1375, XY), in which EGFP driven by the *TFAP2C* promoter (AG) marks hPGCLCs and tdTomato driven by the *DDX4* promoter (VT) marks gonadal stage

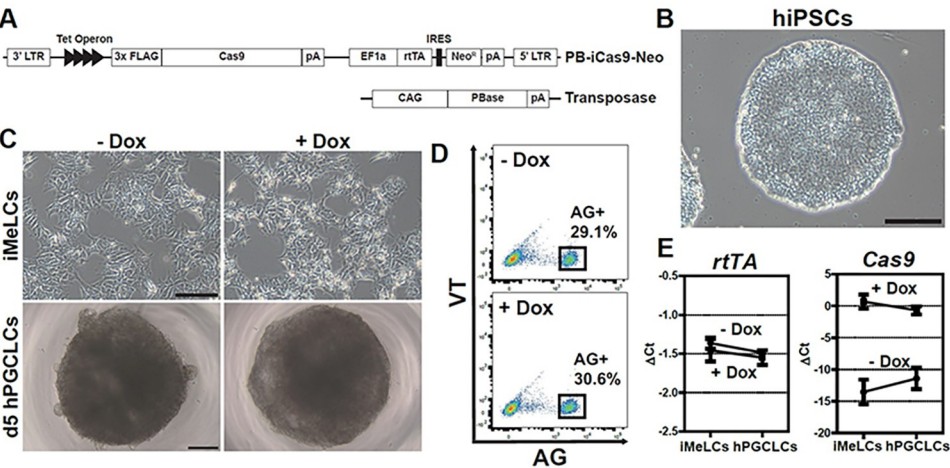

**Fig 1. Establishment of inducible Cas9 (iCas9)-expressing hiPSC line. (A)** Diagram of *piggyBac*-based Cas9 expression under a tet-On inducible system with a PBase (transposase) helper vector. **(B)** A phase-contrast image of 17C-2 (iCas9) hiPSCs. Bar, 200 μm. **(C)** Phase-contrast images of iCas9 hiPSC-derived iMeLCs (top) and day 5 floating aggregates containing hPGCLCs (bottom).–Dox, without Dox; + Dox, treated with 1 μg/ml doxycycline. Bars, 200 μm. **(D)** Fluorescence-activated cell sorting (FACS) analysis of day 5 hPGCLCs derived from iCas9 hiPSCs. AG; TFAP2C-2A-EGFP, VT; DDX4-2A-tdTomato. **(E)** rtTA (top) and Cas9 (bottom) expression during hPGCLC induction, as measured by qPCR. For each gene examined, the ΔCt from the average Ct values of the two independent housekeeping genes *ARBP* and *PPIA* (set as 0) were calculated and plotted. Error bars indicate the standard deviation (SD) of biological replicates.

germ cells [5,9,10]. We isolated seven individual clones and first examined the expression of *rtTA* and *Cas9* by qPCR with or without Doxycycline (Dox) treatment (**S1A Fig**). As expected, *rtTA* was stably expressed in all clones, regardless of Dox concentration. We selected clone 17C-2, which had the highest Dox-inducible expression of *Cas9*, for further characterization prior to CRISPR screening. We confirmed that clone 17C-2 bore Cas9 expression cassettes and heterozygous AGVT alleles (**S1B Fig**). Importantly, 17C-2 hiPSCs could be stably maintained in round and tightly packed colonies with a normal karyotype (**Figs 1B** and **S1C**). We also confirmed that 17C-2 hiPSCs were competent to differentiate into iMeLCs and then AG⁺ hPGCLCs with an induction rate of 38.0% (**S1D** and **S1E Fig**) [5]. We next examined the expression of key marker genes during hPGCLC induction from 17C-2 hiPSCs (**S1F Fig**). As expected, pluripotency markers such as *POU5F1* and *NANOG* were consistently highly expressed in the course of induction, whereas *SOX2* expression was limited to hiPSC and iMeLC stages. *EOMES* was specifically expressed in iMeLC stage, and PGC markers such as *BLIMP1*, *SOX17*, *TFAP2C* and *NANOS3* were largely increased in PGCLCs compared to hiPSCs and iMeLCs.

Prior to undertaking the loss of function CRISPR screen, it was important to determine if Dox treatment affected hPGCLC induction in 17C-2 hiPSCs. Notably, the morphologies of iMeLCs and d5 hPGCLCs (**Fig 1C**) and the induction efficiency of AG⁺ hPGCLCs (29.1% in -Dox vs. 30.6% in +Dox) were comparable in the presence and absence of Dox (**Fig 1D**). *rtTA* expression level was also unaffected by Dox during the induction (**Fig 1E**). Importantly, *Cas9* expression was negligible without Dox, but was highly induced upon Dox treatment in both iMeLCs and d5 hPGCLCs (**Fig 1E**). Thus, we conclude that the inducible *Cas9*-expressing 17C-2 hiPSCs are highly competent to differentiate into germline, despite high levels of Cas9 expression in response to Dox treatment during hPGCLC induction.

## CRISPR screen during the hPGCLC development *in vitro*

To carry out the loss of function CRISPR screen, we next generated 17C-2 hiPSCs containing a custom CRISPR library. Due to technical hurdles in cell culturing, and the difficulty of generating enough cells for high throughput approaches, we opted to screen a focused, sub-genome scale custom sgRNA library. To best enable identification of hPGCLC regulators, we generated a library focusing on genes that were differentially expressed as iPSC differentiated into oogonia-like cells [10]. These 422 coding-genes were targeted with 5 sgRNAs per gene, and 50 non-targeting control sgRNAs were added for a final library size of 2,208 sgRNAs encoded in the pLentiGuide-Puro vector (**S1 Table**).

To optimize the screen, we first determined the lentiviral titer required to achieve a multiplicity of infection (MOI) of 0.3. After 2 days of puromycin selection, we chose 8 μl of lentiviral supernatant per well, which resulted in a 33.3% survival rate [8 μL (+) compared to control, 0 μL (-)], for subsequent CRISPR library establishment (**S2A Fig**). After **17C-2**-derived **c**ustom **C**RISPR library (17C-2-CC) hiPSCs were generated by lentiviral transduction and selection in large scale, we induced hPGCLCs via iMeLCs, to which Dox was added (**Figs 2A** and **S2B**). 17C-2-CC still had a high competency to differentiate into hPGCLCs with a rate of 35.2% of AG$^+$ hPGCLCs at day 5 (**Fig 2B**).

To identify genes potentially driving the hPGCLC state, we isolated Dox-treated AG$^+$ hPGCLCs and AG$^-$ cells from 17C-2-CC after induction and compared them to untreated hiPSCs and iMeLCs samples. In this screen, sgRNAs depleted from the AG$^+$ fraction would represent genes potentially driving the hPGCLC state. Genomic DNA from two biological replicate screens was then harvested and PCR amplicons of sgRNA barcode abundances were quantified via next generation sequencing. Encouragingly, all samples had similar overall read count distributions (**S2C Fig**), and replicates had high global similarity (r = 0.92, 0.76, and 0.86 for correlation between replicates of iMeLCs, AG$^+$, and AG$^-$ samples, respectively) (**S2D Fig**). PCA revealed a conserved trajectory from hiPSCs to iMeLCs to hPGCLCs between replicates (**Fig 2C**). Baseline samples (hiPSCs and iMeLCs) had very low gini indices, suggesting initial uniform library representation, and gini indices approximately doubled in AG$^+$ and AG$^-$ cell fractions (**S2 Table**), consistent with selection of variant populations. The distribution of sgRNA log-fold changes was similar in the AG$^+$ enriched and AG$^-$ enriched sgRNAs (**Fig 2D**), suggesting a roughly equivalent magnitude of selection in both cell populations. We did, however, observe more noise in replicate 1 than 2. Non-targeting sgRNAs, which in a given replicate should not change dramatically in the different cell populations, were strongly correlated in hPGCLC(+) vs. hPGCLC(-) cells as expected in replicate 2, but showed lower correlation in screen 1 (**S3A Fig**). Additionally, among targeting sgRNAs, the distribution of fold changes of all sgRNAs in hPGCLC(+) vs. hPGCLC(-) cells had a significantly greater divergence from normality in replicate 1 than 2 (**S3B Fig**).

We next used the MAGeCK algorithm [11] to call screen hits. 30 genes scored as significantly depleted from the AG$^+$ population at p < 0.01 in replicate 2 (**S3 Table**). Among top hits was *SOX17*, and near-hits included *TFAP2C*, both of which are well-known drivers of the hPGCLC state (**S2E Fig**). Replicate 1 yielded 3 hits, all of which were also hits in replicate 2 (**S3 Table**). This suggests qualitative, though not quantitative, similarities in the screens. To confirm overall biological validity of screen results, we assessed the enrichment of screen hits in RNA-seq data from hPGCLC and iMeLC cells. Merging screen replicates, we found that the set of hits depleted in hPGCLC(+) compared to iMeLC cells were significantly enriched in the list of genes most upregulated in RNA-seq of hPGCLC vs. iMeLC cells (**Fig 2E**). As a control, we found no enrichment of the set of genes enriched in AG$^+$ cells.

To investigate whether the increased noise contributed to the low number of hits in replicate 1, we examined the relation between fold changes and p-values. Each screen replicate had

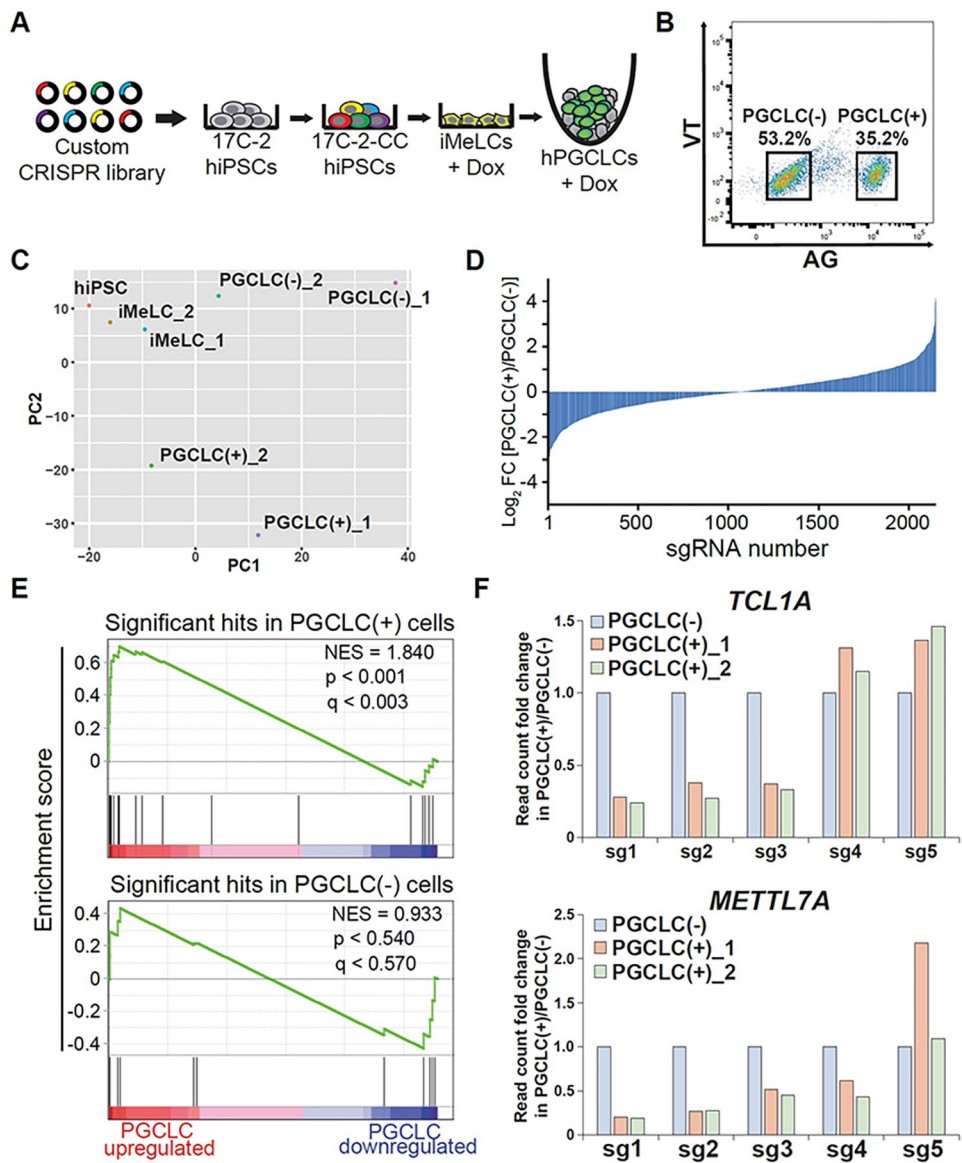

**Fig 2. A CRISPR screen for regulators of hPGCLC development using a custom CRISPR library. (A)** Diagram showing the strategy for CRISPR-based genetic screen during hPGCLC induction *in vitro*. **(B)** FACS sorting of PGCLC(-)/AG$^-$ and PGCLC(+)/AG$^+$ at day 5 of induction. **(C)** PCA of all samples in the screen. **(D)** log$_2$ fold change of all sgRNAs in the custom CRISPR library in PGCLC(+) vs. PGCLC(-) cell populations. Data are average of two replicates. **(E)** Gene sets consisting of the significant hits in PGCLC(+) cells (top) or in PGCLC(-) cells (bottom) tested for enrichment in RNA-seq of genes most upregulated in PGCLC cells. **(F)** Normalized read counts of sgRNAs targeting *TCL1A* (top) and *METTL7A* (bottom) in PGCLC(+) vs. PGCLC(-) populations in screen replicates. Read counts in in PGCLC(-) cells set to 1.0.

16 genes whose sgRNAs had at least a two-fold median depletion from AG$^+$ cells. The average depletion was similar—2.45 in replicate 1 and 2.27 in replicate 2 (**S3 Table**). However, the average p-value of these genes was 0.146 in replicate 1 and 0.006 in replicate 2 (**S3C Fig**). This marked difference in p-values despite similar fold changes indicates that replicate 1 had low statistical power to detect hits due to high variance, explaining the qualitative but not quantitative similarities in the screens. Overall, these results suggest that our screening strategy–which

to our knowledge is the first of its kind in human germline–was effective, and identified a novel set of genes implicated in driving the hPGCLC state.

## Analysis of hPGCLC induction efficiency by *TCL1A* and *METTL7A* knockouts

To identify candidates for further evaluation, we considered genes that scored at p < 0.01 and were in the top 5 in log fold change in either or both screen replicates (**S3 Table**). From this list we selected *TCL1A* and *METTL7A* (**Fig 2F**) for in-depth validation of functionality in hPGCLC specification, as neither has been linked to germ cell function. TCL1A has an oncogenic role in leukemic T-cell [12,13] and can act as an AKT coactivator [14,15]. Its sole MTCP1 domain is encoded by three of its four exons (**S4A Fig**). METTL7A, a putative m[6]RNA methyltransferase [16], has two domains–a signal peptide and a methyltransferase domain–that are encoded by its two exons (**S4B Fig**). To knock out expression of these genes, we decided to remove the ATG start codon and a portion of the domain by two pairs of nCas9 (nickase), respectively (**S4A** and **S4B Fig**). To target each gene, pairs of nickases were cloned into mCherry-expressing vectors and introduced into 1375 (WT) hiPSCs. Two days later, mCherry[high+] cells (2.26% of *TCL1A* KO cells and 1.86% of *METTL7A* KO cells) were isolated via FACS (**S4C Fig**). Single clones were expanded (**S4D Fig**) and screened for the desired deletions via PCR (**S4E Fig**). Clonal *TCL1A* KO (T1) and *METTL7A* KO (M4) cell lines with confirmed deletions by sequencing (**S4F Fig**) were then used for subsequent analysis of hPGCLC induction efficiency.

The *TCL1A* and *METT7A* KO lines were induced into hPGCLCs through iMeLCs (**S4G** and **S4H Fig**) and analyzed by flow cytometry. 5 days after induction, we observed a dramatic reduction in induction efficiency in *TCL1A* KO cells, which yielded 8.46% AG[+] cells (compared to 25.6% WT cells). *METTL7A* KO cells had a more modest, but significant reduction in induction efficiency, generating 17.7% AG[+] cells (**S4I** and **S4J Fig**). Additionally, the number of AG[+] cells per aggregate was reduced in *TCL1A* (526) and *METTL7A* (1,584) KO cells compared to WT (2,850) (**S4K Fig**). Thus, these individual knockout approaches validate our CRISPR screen for regulators of hPGCLC development. Further, these data suggest that loss-of-function of *TCL1A*, and to a more moderate extent *METTL7A*, significantly impairs human germ cell development.

## TCL1A is required for hPGCLC development post-specification

As the phenotype of *TCL1A* KO cells was more severe than that of *METTL7A* KO cells, we focused our analyses on *TCL1A*. In human fetal testes in vivo, *TCL1A* was uniquely expressed in M-prospermatogonia but not in T1-prospermatogonia nor somatic lineages (**Fig 3A**). Concordantly, in our hiPSC-derived male germ cells, *TCL1A* expression is limited to early stage germ cells (AG[+] hPGCLCs, M-prospermatogonia-like cells [MLCs] and transitional cells [TCs]), suggesting the unique role of *TCL1A* in early human germ cells (**Figs 3B, 3C and S5A–S5G**). Therefore, we further investigated the effect of *TCL1A* KO on hPGCLC development by tracing hPGCLC specification and progression over 6 days. *TCL1A* KO hiPSCs and iMeLCs show similar growth to WT counterparts (**Fig 3D**). Furthermore, at day 2, the rate and the number of AG[+] cells per aggregate was similar in KO (17.6%; 1443) and WT (15.5%, 1215) cells (**Fig 3E–3H**). However, at day 4 a significant difference emerged (WT 30.0%, 2176; KO 17.3%, 1191, **P = 0.0012 [% AG[+] cells]**), and at day 6 there was a striking reduction in the number of AG[+] cells per aggregate in *TCL1A* KO cells compared to WT (WT 20.7%, 2843; KO 16.4%, 545, respectively, *P = 0.0230 [AG[+] cells/aggregate]), with *TCL1A* KO cell numbers further decreasing from day 4 to day 6 (**Fig 3E–3I**). This trend suggests that *TCL1A* deletion

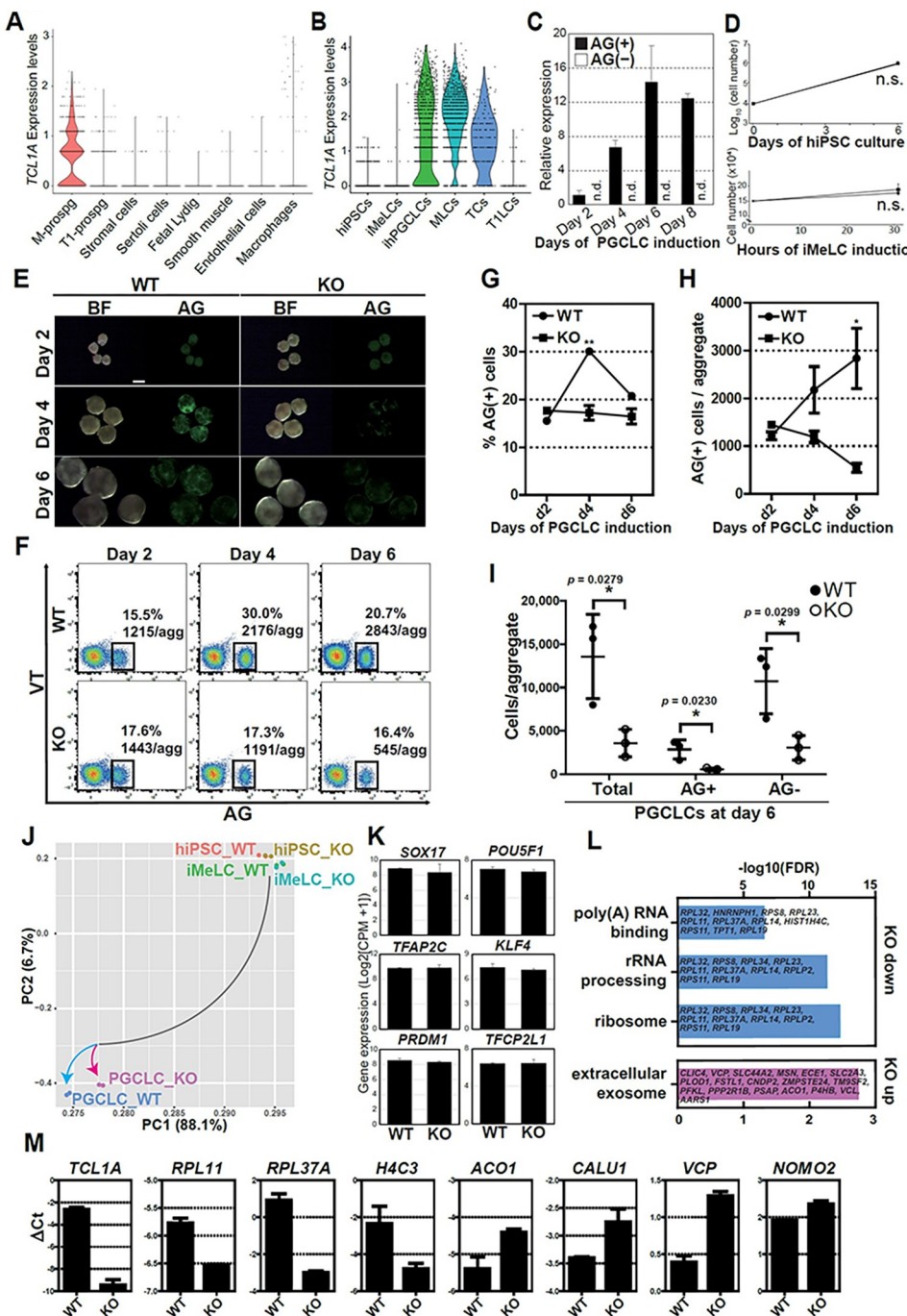

**Fig 3. TCL1A is critical for hPGCLC induction.** (**A, B**) Violin plot showing the expression of TCL1A of the indicated cell types in human fetal testes at 15–16 wpf (A) or hiPSC-derived germ cells (B) [9]. M-prospg, M-prospermatogonia; T1-prospg, T1-prospermatogonia; fetal Lydig, fetal Lydig cells; smooth muscle, smooth muscle cells; MLCs, M-prospermatogonia-like cells; TCs, transitional cells; T1LCs, T1-prospermatogonia-like cells. (**C**) qPCR analysis of *TCL1A* expression of FACS-sorted AG+ and AG− cells at the indicated time points during hPGCLC induction. After calculating ΔCt as in Fig 1E, expression values relative to day 2 value are obtained by $2^{-\Delta\Delta Ct}$ method. Mean ± SD of biological replicates are shown (n = 2). n.d., not detected (Ct values > 30). (**D**) Cell numbers during hiPSC culture (top) and iMeLC induction (bottom) for wild-type (WT, circle) and mutant lines (KO, square). Mean ± SD of three biological replicates. n.s., not significant. (**E**) Bright field (BF) and fluorescence (AG) images of floating aggregates at days 2, 4 and 6 after hPGCLC induction from wild-type (WT, left) or *TCL1A* knockout (KO, right) hiPSCs. (**F**) FACS analysis of AGVT expression during hPGCLC induction in WT (left) and KO lines (right). Boxes denote AG+ cells, and their average percentage and the number of AG+ cells per aggregate from three

independent experiments. **(G, H)** The percentage of AG$^+$ cells **(G)** and the number of AG$^+$ cells per aggregate **(H)** in WT (circles) or KO hPGCLCs (squares) at the indicated time points. Error bars indicate SD of biological triplicates. Statistically significant differences between WT and KO on each day were identified with a two-tailed *t*-test. **$p$ = 0.0012 (% AG$^+$ at d4), *$p$ = 0.0230 (AG$^+$ cells/aggregate at d6). **(I)** The total live cells, AG$^+$ and AG$^-$ cells per aggregate at day 6 of hPGCLC induction are shown. Data represent Mean ± SD of biological triplicates. **(J)** PCA of transcriptomes (two independent experiments) of hiPSCs, iMeLCs and day 4 hPGCLCs in WT and KO lines. Color coding indicates the cell types. **(K)** Gene expression of key germ cell specifier and pluripotency-associated genes retrieved from transcriptome data. Mean ± SD of two biological replicates. **(L)** GO analysis of DEGs between WT and KO hPGCLCs. Representative genes in each GO category are shown. **(M)** Gene expression validation of DEGs between WT and KO hPGCLCs by qPCR. For each gene examined, the ΔCt values from the average Ct values of the two independent housekeeping genes *ARBP* and *PPIA* (set as 0) were calculated and plotted. Error bars indicate SD of biological replicates.

blocks hPGCLC development after specification and results in their gradual loss over time. Similar loss of hPGCLC numbers was observed in another *TCL1A* KO line derived from a different parental hiPSCs, further supporting our finding (**S6A–S6F Fig**).

To further determine how *TCL1A* deletion blocks hPGCLC development, we performed RNA-seq using hiPSCs, iMeLCs and d4 AG$^+$ cells (PGCLCs) and analyzed their global transcriptomes by principal component analysis (PCA) (**Fig 3J**). Consistent with our functional analysis, transcriptomic features in KO hiPSCs and iMeLCs were highly comparable to those of WT. Developmental trajectories starting from hiPSCs and iMeLCs to hPGCLCs undertook a similar course. However, the global transcriptome of PGCLC_KO diverged from that of PGCLC_WT (**Fig 3J**). Key germ cell specifier or pluripotency-associated genes were equally expressed in PGCLC_WT and PGCLC_KO, suggesting that *TCL1A* deletion does not seem to affect germ cell fate specification (**Fig 3K** and **S4 Table**). Based on the profiles, we identified differentially expressed genes (DEGs) in PGCLC_KO relative to PGCLC_WT (FDR-adjusted *p*-value ≤ 0.05). We found 19 down-regulated and 39 up-regulated DEGs in PGCLC_KO (**S4 Table**). Gene ontology (GO) and Kyoto Encyclopedia of Genes and Genomes (KEGG) terms revealed distinct functional classifications for up- and down-regulated genes in PGCLC_KO (**Fig 3L** and **S5** and **S6 Tables**, *p*-value ≤ 0.05). For the down-regulated genes, enriched gene sets were those involved in "Ribosome (hsa03010)", "rRNA processing (GO:0006364)", "poly(A) RNA binding (GO:0044822)" and "protein binding (GO:0005515)" (**Fig 3L** and **S5 Table**). In particular, genes related to Ribosome and rRNA processing, such as seven 60S ribosomal proteins (RPs) (*RPL32, RPL34, RPL23, RPL11, RPL37A, RPL14* and *RPL19*) and two 40S RPs (*RPS8* and *RPS11*) and a ribosomal protein lateral stalk subunit P2 (*RPLP2*), were downregulated in *TCL1A* KO hPGCLCs (**S5 Table**). By contrast, functional GO terms in up-regulated genes included "extracellular exosome (GO:0070062)", "endoplasmic reticulum (GO:0005783)", "ATP binding (GO:0005524)" and "phosphorylation (GO:0016310)" (**Fig 3L** and **S6 Table**). Specifically, *ANKRD13C, VCP, SEC23IP, CALU, ACO1, DHCR24, PLOD1, P4HB, NOMO2* and *ZMPSTE24* associated with endoplasmic reticulum or endoplasmic reticulum membrane were upregulated in *TCL1A* KO hPGCLCs. We also confirmed gene expression changes including down-regulated genes such as *TCL1A, RPL11, RPL37A* and *H4C3*, and up regulation of *ACO1, CALU1, VCP* and *NOMO2* by qPCR (**Fig 3M**). Based on the significant downregulation of ribosome-associated genes, these data support a potential role for *TCL1A*-mediated translational control of hPGCLC development.

## TCL1A regulation of hPGCLC development through AKT signaling

TCL1A is a coactivator of AKT signaling, highlighting a potential role for AKT in hPGCLC development [14,15]. To determine if AKT signaling was affected by *TCL1A* knockout in hPGCLCs, we quantified Pan-AKT and phosphorylated AKT (p-AKT) by

immunocytochemistry (**Fig 4A**), as there was insufficient protein from hPGCLCs for Western blot analysis. All hPGCLCs in both WT and KO were positive for AKT protein and its expression level was unchanged after *TCL1A* knockout (**Fig 4A** and **4C**). However, the intensity of p-AKT was lower in KO ($p = 0.0001$) and the number of p-AKT cells was also reduced in KO (56.5%, 35/62; $p = 0.0018$) compared to WT (27.5%, 17/62) (**Fig 4C** and **4D**). These results indicate that TCL1A acts as an AKT coactivator in hPGCLCs.

Our findings that the percentage and the number of hPGCLCs was decreased by *TCL1A* knockout was associated with downregulation of Ribosome and rRNA processing by RNA-seq, supporting a role for TCL1A in translational regulation of hPGCLC development (**Fig 3L** and **3M**). As the mTOR pathway downstream of AKT signaling regulates cell growth, proliferation, and survival through translational control translational control, we next assessed mTOR signaling by immunocytochemistry (**Fig 4B**). While mTOR protein was globally expressed in hPGCLCs and its expression level was comparable between WT and KO (**Fig 4E**), the expression level of p-mTOR was down-regulated by *TCL1A* knockout (**Fig 4E**; $p = 0.0489$) and the number of p-mTOR positive cells was reduced in KO hPGCLCs compared to WT, although not significantly (**Fig 4F**; $p = 0.0807$). Therefore, mTOR signaling downstream of AKT is altered by *TCL1A* knockout in hPGCLCs.

To determine if defective mTOR signaling following *TCL1A* knockout impacts protein synthesis in hPGCLCs, we next utilized a flow cytometric O-propargyl-puromycin (OP-puro) incorporation assay. After selection of hPGCLCs by surface markers that included PDPN and ITGA6, we found that the peak and level of OP-puro intensity in KO was significantly decreased compared to WT (**Fig 4G and 4H**; $p = 0.0182$). Flow cytometric cell cycle analysis of hPGCLCs by BrdU and 7-AAD incorporation revealed that the proportion of G0/G1 (55.43% to 61.9%) phase was increased, whereas that of S (35.67% to 30.13%) and G2/M (5.51% to 3.44%) phase was decreased in KO relative to WT (**Fig 4I** and **4J**), although only S phase was significantly different ($p = 0.021$). Thus, these results demonstrate that global protein synthesis level and cell cycle were altered through AKT-mTOR signaling pathways in hPGCLCs after *TCL1A* knockout.

## Discussion

Using a novel Dox-inducible Cas9-expressing hiPSC line carrying fluorescent reporters specific for human germline development (iCas9_AGVT_17C-2) (**Fig 1**), we provide the quantitative and qualitative functional analysis of the effect of gene knockout at specific stages of human germ cell development. Notably, this strategy could easily be combined with xrOvary or xrTestis to examine more differentiated cell types [9,10]. Using this novel platform, we conducted the first targeted CRISPR screen designed to identify novel factors important for hPGCLC development (**Fig 2**). Our identification of numerous candidates that could potentially contribute to human germ cell development, one of which is described further below, provide essential insight into hPGCLC development and identify multiple pathways for future mechanistic studies.

Although key transcription factors for human germline specification, such as SOX17 [4,17], BLIMP1[5], and TFAP2C [17–19] have been functionally assessed, the mechanistic basis for hPGCLC-specification remains largely enigmatic. Among candidates identified by CRISPR screen analysis in the current work, we showed that TCL1A plays a role in hPGCLCs development post-specification (**Fig 3**). Although induction of hPGCLCs progressed normally in *TCL1A* KOs, the number of AG+ cells decreased over time (**Fig 3E–3I**). Thus, TCL1A appears important for the proliferation and/or survival of cells already committed to the germline, thus differing from well-known key elements contributing to the early specification of hPGCLCs

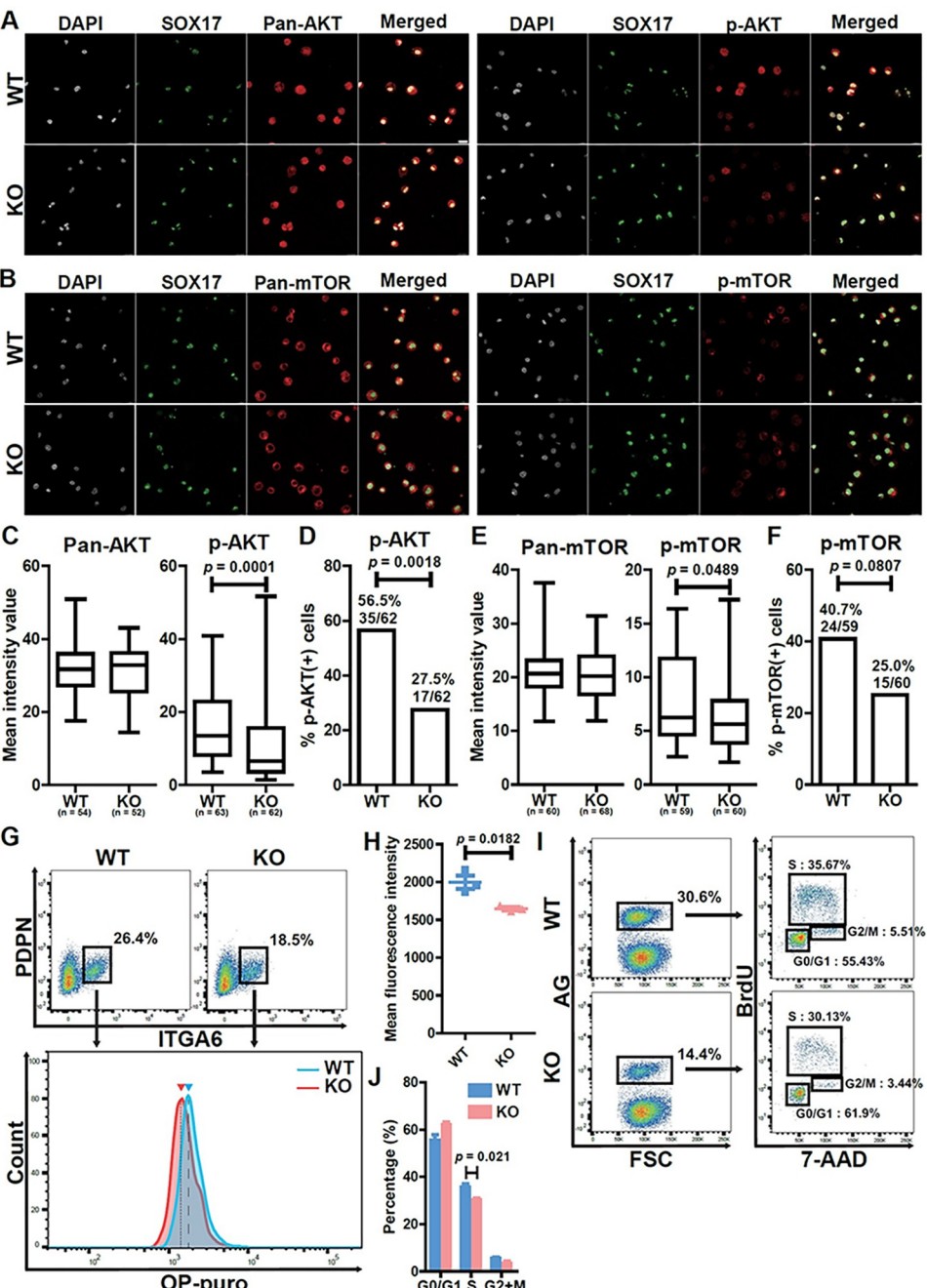

**Fig 4. Functional analysis of TCL1A in hPGCLCs. (A, B)** Immunocytochemistry analysis of 1375 (WT) and *TCL1A* knockout (KO) hPGCLCs for DAPI (white), SOX17 (green), pan-AKT, phospho (p)-AKT, pan-mTOR or p-mTOR (red). Bar, 20 μm. **(C, E)** Box plot quantification of the mean intensity values of the indicated markers, Center line, median; box limits, upper and lower quartiles; whiskers, 1.5× interquartile range. Statistically significant differences between WT and KO were identified with Mann–Whitney *U* test. *P* = 0.0001 (p-AKT), 0.0489 (p-mTOR). n, number of SOX17[+] cells counted. **(D, F)** Bar graphs showing the proportions of the indicated phosphorylated proteins in SOX17[+] cells. Statistically significant differences were identified with Fisher's exact test. *P* = 0.0018 (p-AKT), 0.0807 (p-mTOR) **(G)** Protein synthesis assay in WT and KO hPGCLCs, on the basis of OP-puro incorporation. (top) FACS analysis of ITGA6 and PDPN expression. Boxed areas indicate the percentage of ITGA6[+]PDPN[+] hPGCLCs. (bottom) Representative FACS histograms of OP-puro incorporation in ITGA6[+]PDPN[+] hPGCLCs derived from WT or KO lines. The blue arrowhead and dashed line indicate the mean fluorescence intensity (MFI) of WT hPGCLCs, and the red arrowhead and dotted line indicate the MFI of KO hPGCLCs. **(H)** Scatter dot plot of MFI of OP-puro, measured by FACS. Each dot (blue, WT; red, KO) represents three independent experiments, and lines indicate mean MFI values. Statistically significant differences were identified with two-tailed *t*-test. *P* = 0.0182. Error bars indicate SD of

biological triplicates. **(I)** Cell cycle analysis of WT and KO hPGCLCs by BrdU and 7-AAD corporation. (left) FACS analysis of AG expression. The percentages of AG$^+$ cells (highlighted in boxes) are shown. (right) BrdU and 7-AAD analysis of AG$^+$ cells. The average percentage of cells in the indicated cell cycle stages are shown. **(J)** Bar graph of the percentage of each cell cycle stage. Statistically significant differences were identified with two-tailed *t*-test. *P* = 0.021 (S phase). Error bars indicate SD of biological triplicates.

[5,17,18]. Of note, among cells within day 4 PGCLC aggregates, AG$^−$ cells were also affected in *TCL1A* KO compared to WT, although this reached statistical significance in only one of the mutant lines (**Figs 3I** and **S6F**). This could be due to the functional role in *TCL1A* in AG$^−$ cells, or indirect influence of AG$^+$ hPGCLCs on AG$^−$ cell growth, which warrant further investigation.

*TCL1A* is a proto-oncogene first identified in leukemic T-cells that acts as a coactivator of AKT signaling [13–15,20]. In primordial germ cells, AKT appears to be activated by kit ligand (also known as stem cell factor, SCF)-dependent PI3K signaling [21,22]. As shown in **Fig 4A, 4C and 4D**, phosphorylation of AKT was decreased in *TCL1A* KO hPGCLCs. Notably, transcriptome analysis of WT and KO hPGCLCs revealed that TCL1A-AKT signaling likely impacted translation, as GO and KEGG analysis revealed that terms related to translation such as rRNA processing and Ribosome were down-regulated in KO cells while terms related to endoplasmic reticulum were up-regulated (**Fig 3L**). Interestingly, previous studies revealed that TCL1A was significantly decreased during hPGCLC expansion culture, suggesting that additional metabolic changes contribute to human germ cell proliferation [23].

mTOR signaling downstream of AKT mediates translational control [24], driving global expression of most RP genes [25,26] and modulating the activity of transcription factors, such as TIF-1A, important for Pol I activity and rRNA synthesis [27]. Consequently, the AKT-mTOR signaling pathway drives ribosome biogenesis and global protein synthesis critical for cell growth, proliferation, and survival. Consistent with these effects, we found that TCL1A-mediated AKT signaling is compromised in hPGCLC KOs and results in downregulation of mTOR signaling (**Fig 4B, 4E** and **4F**). Moreover, in hPGCLCs, inhibition of mTOR signaling by mutant TCL1A reduced global level of protein synthesis, likely contributing to decreased cell proliferation and/or survival (**Fig 4G–4J**). Thus, TCL1A-dependent regulation of AKT-mTOR signaling appears to play a critical role in translational control in hPGCLCs, ultimately leading to their decrease in number over time following knockout. Translation of a subset of mRNAs responsible for hPGCLCs development appears to be controlled by mTOR signaling in other cell types as well [28,29], further highlighting the importance of this pathway.

There are certain limitations to the approach taken. High content CRISPR screening has become a common, powerful forward genetics method that is highly effective in cell lines. However, CRISPR screening in primary cells with more heterogeneity, lower numbers of cells, and sensitive culturing conditions is more challenging. Further, cell fate-based screens, which often rely on surface markers and are sensitive to gating schemes, are more inherently challenging than proliferation/viability screens. Accordingly, it was not surprising that a degree of noise was observed in our screens and that some, but not all, positive control genes scored as hits. Ultimately, as with many screens performed in challenging contexts, it must be assumed that some hits will be false negatives. However, the detection of confirmation of true positive hits is evidence of screen success overall.

In conclusion, we have generated essential genetic resources critical for further mechanistic study of hPGCLC development and developed a CRISPR screening strategy that will aid in the further identification of genes involved in human germline development. Using this platform, our current study demonstrated the potential importance of TCL1A regulation of AKT-mTOR dependent translational control of cell proliferation and/or survival in hPGCLCs. As

translational control of germline stem cells of *Drosophila* [30] has previously been documented, our studies clearly support further studies on the translational regulation of human germ cell development and reproduction.

## Materials and methods

### Culture of hiPSCs

hiPSCs were cultured on plates coated with recombinant laminin-511 E8 (iMatrix-511 Silk, Nacalai USA) and maintained under feeder-free conditions in StemFit Basic04 medium (Ajinomoto) containing 20 ng/ml basic FGF (Peprotech) at 37˚C under a 5% $CO_2$ atmosphere. Before passaging or the induction of differentiation, hiPSC cultures were treated with a 1:1 mixture of TrypLE Select (Life Technologies) and 0.5 mM EDTA/PBS for 15 min at 37˚C to dissociate them into single cells. Subsequently, 10 μM ROCK inhibitor (Y-27632; Tocris) was added in culture medium for 1 day after passaging hiPSCs.

### Generation of AGVT-knock-in reporter hiPSCs (AGVT 14C10 hiPSCs)

The donor vector and the TALEN constructs for generating the *TFAP2C/AP2γ-p2A-EGFP* (AG); *DDX4/hVH-p2A-tdTomato* (VT) alleles were described previously [5,9,10]. Homology arms for *TFAP2C* and intervening *2A-EGFP-loxP* sequence were amplified from genomic DNA derived from 585B1 898 hiPSCs [5], and subcloned into pCR2.1 vector using the TOPO TA cloning kit (Life Technologies). *MC1-DTA-pA* cassette synthesized by Gene Universal (Newark, DE) was inserted downstream of the right homology arm using *Kpnl* restriction sites (designated as YS42). The *p2A-EGFP* fragment with the *PGK-puro* cassette flanked by the *loxP* site was also PCR-amplified using the donor vector previously used to generate WT1-p2A-EGFP knock-in hiPSCs [31], and inserted in-frame at the 3'-end of the TFAP2C coding region obtained from YS42 through inverse PCR. TALEN's RVD sequences were as follows: TFAP2C-left (5-prime), NN NN NI NN NI NI NI HD NI HD NI NN NN NI NI NI NG; TFAP2C-right (3-prime), NI HD NG HD NG HD HD NG NI NI HD HD NG NG NG HD NG. DDX4-left, HD HD HD NI NI NG HD HD NI NN NG NI NN NI NG NN NI NG NN; DDX4-right, NN NI NI NN NN NI NG NN NG NG NG NG NN NN HD NG NG. To establish AGVT-knock-in reporter hiPSCs, we first transfected the donor (5 ug) and TALEN-expression vectors (2.5 ug each) for AG into one million hiPSCs (clone 1390G3 derived from peripheral blood mononuclear cells of female donor, kind gift from Dr. Nakagawa at Kyoto University [32]) by NEPA21 Type II Electroporator (Nepagene), followed by puromycin selection. Puromycin-resistant colonies were pooled, expanded then subsequently transfected by donor and TALEN-expression vectors targeting VT as described above, followed by neomycin selection and transfection of a plasmid expressing Cre recombinase to remove the *PGK-Puro* and *PGK-Neo* cassettes. The success of the targeting, random integration and Cre recombination process were confirmed by PCR no extracted genomic DNA from each colony using primer pairs listed in S7 Table. One clone bearing homozygous AG and VT alleles (14C10), along with 1375 hiPSCs were used for further targeting *TCL1A* loci as described below.

### Generation of inducible Cas9 (iCas9) expressing hiPSC lines

For the construction of an inducible Cas9 expression piggyBac vector (PB-iCas9-Neo), a Gateway entry vector containing Cas9, pENTR-hSpCas9, was cloned into a PB-TA-ERN plasmid (all-in-one piggyBac transposon destination vector for doxycycline [dox]-inducible expression with constitutive rtTA and neomycin resistance [33,34]) with Gateway LR Clonase II enzyme mix (Invitrogen). The reactions were performed according to the manufacturer's protocol.

These plasmids were provided by Dr. Knut Woltjen at Kyoto University. The PB-iCas9-Neo vector (2 μg) and pCAG-PBase vector (1 μg) (transposase expression vector, a gift from Dr. Knut Woltjen [35]) were introduced into 1 million AGVT hiPSCs (585B1 1375, gift from Dr. Mitinori Saitou, Kyoto University) by electroporation with a NEPA21 Type II electroporator (Nepagene). Single colonies were isolated after selection with neomycin and chosen for expansion and characterization.

For the measurement of rtTA and inducible Cas9 expression, each clone was treated with Dox (Takara) at 0, 250 or 1000 ng/ml for 2 days. Total RNA was extracted with an RNeasy Micro Kit (QIAGEN) according to the manufacturer's instructions. Total RNA (1 μg) was used as a template for cDNA synthesis with the SuperScript III First-Strand Synthesis System (Invitrogen). The cDNA was serially diluted five-fold, and equal amounts were PCR amplified. Q-PCR was performed with Power SYBR Green PCR Master mix (Life Technologies) and a StepOnePlus Real-Time PCR System (Applied Biosystems). The gene expression levels were examined through calculation of ΔCt (on log2 scale) normalized to the average ΔCt values of *PPIA* and *ARBP*, with the primer pairs listed in **S7 Table**. The success of the integration of the Cas9 cassette was assessed by PCR of the extracted genomic DNA from each clone. Clone 17C-2 (designated as iCas9_AGVT_17C-2) was selected for iCas9 hiPSCs.

## Karyotyping and G-band analyses

hiPSCs were incubated in culture medium containing 100 ng/ml of KaryoMAX Colcemid solution (Gibco) for 8 h. After dissociation by TrypLE Select, cells were treated with pre-warmed buffered hypotonic solution and incubated for 30 min at 37°C. Cells were then fixed with Carnoy's solution (3:1 mixture of methanol and acetic acid) and added dropwise onto glass slides for preparation of chromosomal spreads. Karyotypes were first screened by counting the numbers of chromosomes identified by DAPI staining. Cell lines bearing 46 chromosomes were further analyzed on the basis of G-banding by Cell Line Genetics (Madison, WI).

## Induction of hPGCLCs

hPGCLCs were induced from hiPSCs via iMeLCs as described previously [5]. For the induction of iMeLCs, hiPSCs were plated at a density of $4\times10^4$ to $5\times10^4$ cells/cm$^2$ onto a human fibronectin (Millipore)-coated 12-well plate in GK15 medium (Life Technologies) with 15% KSR, 0.1 mM NEAA, 2 mM L-glutamine, 1 mM sodium pyruvate and 0.1 mM 2-mercaptoethanol) containing 50 ng/ml of ACTA (R&D Systems), 3 μM CHIR99021 (Tocris Bioscience) and 10 μM of a ROCK inhibitor. After 31 h, iMeLCs were harvested and dissociated into single cells with TrypLE Select. For induction of hPGCLCs, iMeLCs were then plated at 3,500 cells per well into a low-cell-binding V-bottom 96-well plate (Thermo Fisher Scientific) containing GK15 medium supplemented with 200 ng/ml BMP4 (R&D Systems, 314-BP-010), 100 ng/ml SCF (R&D Systems, 255-SC-010), 50 ng/ml EGF (R&D Systems, 236-EG), 10 ng/ml LIF (Millipore, #LIF1010) and 10 μM of Y-27632 (GK15+BSELY). The plates were incubated at 37°C under a 5% $CO_2$ atmosphere until use in downstream assays.

## Q-PCR analysis of marker expression

Total RNA from iMeLCs and hPGCLCs was extracted with a RNeasy Micro Kit (QIAGEN) according to the manufacturer's instructions. The concentration of total RNA was measured with a Qubit 3 fluorometer (Invitrogen). Synthesis and amplification of cDNAs with 1 ng of purified total RNA was performed as previously described [36]. Q-PCR on amplified cDNA was performed with Power SYBR Green PCR Master mix (Life Technologies) and a StepOnePlus Real-Time PCR System (Applied Biosystems). The gene expression levels were examined

by calculation of $\Delta C_t$ (on $\log_2$ scale) normalized to the average $\Delta C_t$ values of *PPIA* and *ARBP*, with the primer pairs listed in **S7 Table**.

## Fluorescence-activated cell sorting (FACS)

We isolated AG[+] hPGCLCs from floating aggregates by using FACS. Floating aggregates containing hPGCLCs, which were induced from hiPSCs, were dissociated into single cells through 0.1% trypsin/EDTA treatment for 15 min at 37˚C with periodic pipetting. After the reaction was quenched by addition of an equal volume of fetal bovine serum, cells were resuspended in FACS buffer (0.1% BSA in PBS) and strained through a 70 μm nylon cell strainer (Thermo Fisher Scientific) to remove cell clumps. AG[+] cells were sorted with a FACSAria Fusion flow cytometer (BD Biosciences) and collected in an microcentrifuge tube containing CELLOTION (Amsbio). All FACS data were collected in FACSDiva v8.0.2 software (BD Biosciences) and analyzed with FlowJo v10.8.1 (BD Biosciences). For flow cytometry analysis, hPGCLCs were dissociated and resuspended in FACS buffer. AG[+] cells were analyzed with FACSCanto or FACSFortessa (BD Biosciences).

## CRISPR/Cas9 Screen analysis

A total of 422 genes were selected from 453 previously reported genes variably expressed during the transition of hPGCLC-derived cells in xenogeneic reconstituted ovaries, after removal of pseudogenes and long-non-coding RNAs [10]. A custom CRISPR gRNA library pool containing 2159 gRNAs was designed and synthesized by GenScript (Piscataway, NJ). The PCR products were amplified and ligated into the pLentiGuid-Puro vector through *BsmBI* restriction via GenBuilder[TM] Plus assembly. The library was verified through next generation sequencing by GenScript. For lentivirus packaging, the target vector and pCMV-VSV-G and pCMV-dR8.2 dvpr lentiviral packaging plasmids (Addgene plasmid #8454 and #8455, gifts from Dr. Robert Weinberg, Massachusetts Institute for Technology) were co-transfected into 293T cells (Clontech, 632180) with PEI reagent (Polysciences, 23966–1). Lentiviral particles were collected 48 h and 72 h after transfection and filtered.

For CRISPR screening, 1 million 17C-2 hiPSCs were transduced with 0, 8 or 16 μl of lentivirus supernatant for 6 h in a 6 cm dish. One day after transduction, cells were treated with puromycin for 48 h. Three days after transduction, hiPSCs transduced with each titer were harvested and counted, and an MOI of 0.3 was measured (**S2A Fig**). The transduction with 8 μl of lentivirus supernatant was identified as the optimal titer to achieve an MOI of 0.3. Three million 17C-2 hiPSCs in three 6 cm dishes were then transduced with 8 μl lentivirus supernatant per dish, selected with puromycin and expanded in bulk; the gRNA fold coverage was maintained, and these cells were designated as 17C-2-CC hiPSCs. iMeLCs and hPGCLCs derived from 17C-2-CC were induced as described above, except that 1 μg/ml Dox was added for induction of Cas9 expression. In every step, the gRNA fold coverage was maintained above 200×. At day 5, floating hPGCLC aggregates, and AG[+] [designated as PGCLC(+)] and AG[-] cells [designated as PGCLC(-)] were sorted with a FACSAria Fusion flow cytometer and collected in 15 ml tubes containing CELLOTION. Genomic DNA from hiPSCs, iMeLCs, PGCLC (+) and PGCLC (-) was isolated with a GenElute Mammalian Genomic DNA Miniprep kit (Sigma).

Genomic DNA libraries for sequencing were then prepared according to previously described protocols (Joung *et al*, 2017) maintaining >200X fold coverage. Briefly, PCR-amplified library samples were purified with the QIAquick PCR Purification Kit (Qiagen) followed by gel extraction with the QIAquick Gel Extraction Kit (Qiagen). The barcoded libraries were then pooled at equimolar ratios and sequenced on a NextSeq500/550 instrument (Illumina,

150 cycles High Output kit v2.0) to generate 150-bp single-end reads. MAGeCK software was used for screening analysis [11] using default settings to call hits in PGCLC(+) vs. PGCLC(-) cells, with screen replicates analyzed separately.

## Generation of *TCL1A* and *METTL7A* knockout hiPSC lines

Two pairs of sgRNA sequences before and after the targeted exons of *TCL1A* and *METTL7A* were designed with the Molecular Biology CRISPR design tool (Benchling). Two pairs of sgRNAs targeting *TCL1A* (CGAGTGCCCGACACTCGGGG and GCAAGAGCCAGAGCC TCTCA for pre-exon1, and AGGTACAGCCAGCTTTGGAG and GGCTGTACCTCGATG GTTAA for post-exon1) or *METTL7A* (TACCATCTTTATCCTGAGAC and GTAAGCTCC ATTGCTCAGAA for pre-exon1, and CGGGGAGAACATGCACCAGG and AGCGCTCA AACTGCAGGTGT for post-exon1) were selected and cloned into the pX335-U6-Chimeric BB-CBh-hSpCas9n (D10A) SpCas9n-expression vector to generate the sgRNAs/Cas9n vector (Addgene, #42335 [37]). Four sgRNAs/nCas9 vectors (2.5 μg each) were introduced into 1 million AGVT hiPSCs (1375 and 14C10) by electroporation with a NEPA21 Type II electroporator. At 2 days after transfection, mCherry-expressing cells were sorted into a 96-well plate (single cell/well) with a FACSAria Fusion flow cytometer, and subsequently expanded and genotyped by PCR (S4E Fig). Large deletions in T1 and M4 were further validated by Sanger sequencing (S4F Fig).

## RNA-seq Analysis

Total RNA from hiPSCs, iMeLCs and hPGCLCs (AG$^+$) was extracted with an RNeasy Micro Kit, and RNA quality was evaluated with High Sensitivity RNA Screen Tape on an Agilent 4200 TapeStation. The cDNA library was prepared with a Takara SMART-Seq HT kit, and adapters were ligated with a Nextera XT DNA Library Preparation Kit and Index Kit v2 according to the manufacturers' protocols. AMPure XP beads were used for primer cleanup. Subsequently, 75-base pair reads were sequenced on a NextSeq 500 instrument (Illumina, San Diego, CA) according to the manufacturer's protocol.

The quality of FASTQ files was verified with FastQC v0.11.2. All samples were aligned to the human genome, build hg38, with Rsubead aligner v1.24.1 [38]. The number of fragments overlapping each Entrez gene were summarized by using featureCounts [39] and Rsubread's inbuilt hg38 annotation. Differentially expressed genes (DEGs) were identified with the DESeq2 package with FDR $\leq$ 0.05 as the significance threshold. DEGs were mapped to GO terms by using DAVID (v6.8) with the background list set to "Homo sapiens" [40]. Only enriched GO terms with $p < 0.05$ were shown.

## Mapping reads and data analyses for single cell RNA-sequencing (scRNA-seq)

scRNA-seq files for 9A13 hiPSC-derived cells (hiPSCs, iMeLCs, hPGCLCs_1, hPGCLCs_2, d81_xrTestis, d124 xrTestis) and three human fetal testes (Hs26-18W0d [16 wpf], HS27-18Wd5 [16 wpf], Hs31-17W3d [15 wpf]) were retrieved from NCBI GEO (GSE153819). Primary and secondary analyses were performed as described previously [9]. First, raw data were demultiplexed with the mkfastq command in cellranger (V5.0.1) to generate Fastq files. Trimmed sequence files were mapped to the reference genome for humans (GRCh38) provided by 10x Genomics. Read counts were obtained from outputs from Cell Ranger. Secondary data analyses were performed in R (v.4.1.0) with Seurat (v.4.0). UMI count tables were first loaded into R using the Read10X function, and Seurat objects were built from each sample. Cells with fewer than 200 genes, an aberrantly high gene count above 8000, or a percentage of

total mitochondrial genes >15% were filtered out. Samples were combined, and the effects of mitochondrial genes and cell cycle genes were regressed out with SCTransform during normalization in Seurat; then, gene counts were scaled by 10000 and natural log normalized. Mitochondrial genes and cell cycle genes were excluded during cell clustering and dimensional reduction. Cells were clustered according to Seurat's shared nearest neighbor modularity optimization-based clustering algorithm. Clusters were annotated based on previously characterized marker gene expression, and cluster annotation was generated for downstream analyses. Dimensional reduction was performed with the top 3000 highly variable genes and the first 30 principal components with Seurat. Data were visualized using ggplot2.

## Immunofluorescence (IF) analysis

For IF staining of hPGCLCs, day 4 aggregates were dissociated into single cells and spread onto poly-L-lysine-treated Superfrost Plus Microscope Slides (Fisher Scientific). The slides were fixed in 4% paraformaldehyde in PBS for 15 min at room temperature (RT), washed three times with PBS for 5 min, permeabilized with 0.2% Triton-X in PBS for 10 min at RT and washed three times with PBS for 5 min. Next, the slides were incubated with primary antibodies to pan-AKT (CST, 4691T), phospho-AKT (Ser473; CST, 4060T), pan-mTOR (CST, 2983T) and phospho-mTOR (Ser2448; CST, 5536T) in blocking solution (1% bovine serum albumin in PBS) overnight at 4˚C. After being washed three times with blocking solution for 5 min, the slides were incubated with secondary antibodies in blocking solution and 1 μg/ml of DAPI for 1 hr in the dark at RT. The slides were washed with blocking solution three times for 5 min, then mounted in Vectashield mounting medium (Vector Laboratories) for confocal laser scanning microscopy analysis (Leica, SP5-FLIM inverted). Confocal images were processed in Leica LasX (v3.7.2). The mean fluorescence intensity of each marker in SOX17-positive cells was measured in ImageJ software (National Institutes of Health, Bethesda, MD).

## Cell cycle analysis by BrdU and 7-AAD incorporation

BrdU (1 mM) in GK15 was diluted in GK15+BSELY medium to obtain a final concentration of 20 μM. Then half the medium of hPGCLC cultures in V-bottom plates was changed (final BrdU concentration of 10 μM). BrdU-treated aggregates were incubated at 37˚C under 5% $CO_2$ in air for 1 h. hPGCLCs were dissociated into single cells, and cell cycle analysis was performed by staining cells with BrdU-APC and 7-AAD with BD Pharmingen APC BrdU Flow Kits (552598, BD Biosciences), according to the manufacturer's instructions. Stained cells were analyzed in FACSFortessa.

## Protein synthesis assay by OP-puro incorporation

O-propargyl-puromycin (OP-puro) labeling was performed with a Protein Synthesis Assay Kit (ab235634; Abcam) according to the manufacturer's instructions. Protein Label stock (200×) was diluted in GK15+BSELY medium to produce 1× Protein Label solution. Then day 4 aggregates containing hPGCLCs were transferred onto Nunc 4-Well Dishes (Thermo Fisher Scientific) containing 1× Protein Label solution. Dishes were incubated at 37˚C under 5% $CO_2$ in air for 1 h. To identify hPGCLCs, we used surface markers, because GFP signals would have been quenched by the sodium azide treatment during the procedure. hPGCLCs were dissociated into single cells and then stained with Alexa Fluor 647-conjugated rat anti-human PDPN (337007, BioLegend) and BV421-conjugated anti-CD49f (INTEGRINα6) (313623, BioLegend) in FACS buffer for 30 min at RT before fixation. Protein synthesis was detected with FACSFortessa.

## Supporting information

**S1 Fig. Generation and selection of iCas9 hiPSC clones, associated with Fig 1. (A)** Gene expression of *rtTA* (left) and *Cas9* (right) in the indicated hiPSC clones. White, gray and black bars indicate cells treated with Dox at 0, 250 or 1000 ng/ml, respectively. The quantification of the gene expression levels was as shown in Fig 1E. Error bars indicate SD of technical duplicates. Clone 17C-2 (highlighted in red) was selected for downstream CRISPR screening assays because it showed the highest *Cas9* expression after Dox treatment. **(B)** PCR genotyping of the *TFAP2C-2A-EGFP* (AG) (left), *DDX4-2A-tdTomato* (VT) (middle) and *Cas9* coding sequence (right). Rec., recombined with fluorescent protein; non, non-targeted; yellow arrow, 3 kb; yellow asterisk, 1 kb. **(C)** Representative results of 17C-2 hiPSC karyotype analysis, showing a normal karyotype (46, XY). **(D)** Phase-contrast image of 17C-2 hiPSC-derived iMeLCs. Bar, 200 μm. **(E)** Bright-field image of a day 5 floating aggregate derived from 17C-2 hiPSCs (top) and its FACS plot. Bar, 200 μm. The percentages of AG$^+$ cells (highlighted in box) are shown. **(F)** Gene expression dynamics of key markers during hPGCLC induction from iCas9 hiPSCs. hPGCLCs were harvested on day 5. Error bars indicate the standard deviation (SD) of technical duplicates. n.d., not detected.
(PDF)

**S2 Fig. Generation of 17C-2 iCas9 hiPSCs containing a custom CRISPR library (17C-2-CC) by lentiviral transduction, associated with Fig 2. (A)** (left) Phase-contrast images of lenti-viral transduced 17C-2 hiPSCs before (day 1) and after (day 3) puromycin selection. Volumes of virus-containing supernatant added per well (0 μl, 8 μl and 16 μl; virus titer) are indicated. (+), with puromycin; (-), without puromycin. (right) The number of cells counted at day 3 in each titer. 0 (-) is set as 100%. Titer 8 was selected as the optimized titer for a multiplicity of infection (MOI) of 0.3. **(B)** Phase-contrast images of 17C-2-CC hiPSCs (left), iMeLCs (middle) or day 5 floating aggregates containing hPGCLCs derived from 17C-2-CC hiPSCs (right). Bars, 200 μm. **(C)** Distribution of abundances of normalized sgRNA read counts in all screen samples. **(D)** Comparison of read counts of all sgRNAs in biological replicates of PGCLC(+) screen samples. **(E)** Normalized read counts of sgRNAs targeting *TFAP2C* (left) and *SOX17* (right) in PGCLC(+) vs. PGCLC(-) populations in screen replicates. Read counts in in PGCLC(-) cells set to 1.0.
(PDF)

**S3 Fig. Analysis of variance and statistical power of screen replicates 1 and 2. (A)** Scatterplots showing correlation between non-targeting sgRNAs in hPGCLC(+) vs. hPGCLC(-) in screen replicates 1 (top) and 2 (bottom). r = Pearson correlation coefficient. **(B)** Distribution of fold changes in hPGCLC(+)/hPGCLC(-) cells in replicates 1 (left) and 2 (right). Deviation from normal distribution quantified by skewness and kurtosis. **(C)** -log$_2$ p-values of genes depleted more than two-fold in hPGCLC(+) vs. hPGCLC(-) in replicates 1 (left) and 2 (right).
(PDF)

**S4 Fig. Knockout scheme of *TCL1A* and *METTL7A* mutant hiPSCs and analysis of their induction efficiency of hPGCLCs, associated with Fig 3. (A, B)** Knockout schemes for human *TCL1A* (A) or *METTL7A* (B) loci. I in A, TCL1_MTCP1 domain; I and II in B, METTL7A signal peptide and Methyltransf_11 domain, respectively. Black boxes indicate exons. Red arrowheads indicate target sites recognized by the pair of Cas9 nickases flanking the start codons and the domain-encoding regions. Arrows indicate primer sites for genotyping. **(C)** FACS sorting of 1375 hiPSCs (WT) on the basis of mCherry expression by px335-derived nickase vectors targeting *TCL1A* (left) and *METTL7A* (right). The percentage of mCherry$^{high}$ cells (highlighted in boxes) and the sorting gates are shown. **(D)** Phase-contrast images of

single clonal expansion of *TCL1A*- and *METTL7A*-targeted hiPSCs. Bar, 200 μm. (**E**) PCR genotyping of the large deletion in *TCL1A* (left) and *METTL7A* (right) exons by double pairs of nickase. Yellow asterisk, 500 bp. T1 and M4 (red) clones were selected for further analysis of each knockout line. (**F**) Sanger sequencing results of large deletions in T1 (left) and M4 (right). The thin black line represents a gap between target sites recognized by the pair of Cas9 nickases. (**G, H**) Phase-contrast images of iMeLCs (**G**) and day 5 floating aggregates containing hPGCLCs (**H**) derived from T1 (left) and M4 (right). Bars, 200 μm. (**I**) FACS analysis of day 5 hPGCLCs derived from WT (left), T1 (middle) and M4 (right). Boxes indicate AG$^+$ cells. The average percentage and number of AG$^+$ cells per aggregate from two independent experiments are also shown. (**J, K**) Percentage of AG$^+$ cells (**J**) and number of AG$^+$ cells per aggregate (**K**) in WT (white), T1 (gray) and M4 (black) in day 5 hPGCLCs. Error bars indicate SD of biological replicates.
(PDF)

**S5 Fig. Annotation of cell types in single cell (sc)RNA-seq data of human fetal testes and hiPSC-derived germ cells, associated with Fig 3.** (**A**) tSNE plot showing different cell types in human fetal testes at 15–16 week post fertilization (wpf) [9]. Cell types were assigned based on the known markers as shown in B and C and colored accordingly. (**B, C**) Expression of key marker genes associated with indicated cell types [9]. (**D, E**) scRNA-seq data of hiPSC-derived cells during hPGCLC induction originated from three samples (hiPSCs, iMeLCs, hPGCLCs) were projected on tSNE and annotated based on the key markers as shown in (E). (**F, G**) scRNA-seq data of hiPSC-derived germ cells obtained from xenogeneic reconstituted testes (xrTestes) were annotated based on key markers as shown in (**G**).
(PDF)

**S6 Fig. Induction of hPGCLCs from another TCL1A mutant hiPSCs, associated with Fig 3.** (**A**) Genotyping PCR to screen for clones bearing *TFAP2C-p2A-EGFP* (AG) (top) and *DDX4-p2A-tdTomato* (VT) alleles (bottom). Note that clone 10 (designated as 14C10 hiPSCs) bear biallelic AG and VT alleles. 585B1 1375 hiPSCs (monoalleleic for both AG and VT alleles) were used as positive control. (**B**) Genotyping PCR to screen for *TCL1A* mutant clones. Clone 9 (designated as 6C9 hiPSCs) was used for downstream assays. 14C10 hiPSCs were included as positive control. (**C**) Sequencing chromatogram of 6C9 hiPSCs at *TCL1A* loci showing bialleleic deletion (134 and 149 bp). (**D**) Representative results of karyotype analysis for 14C10 and 6C9 showing normal female karyotypes. (**E**) (top) Bright field (BF) and fluorescence (AG) images of floating aggregates at day 5 after hPGCLC induction from wild-type (WT, 14C10, left) or *TCL1A* KO (KO, 6C9, right) hiPSCs. (bottom) FACS analysis of day 5 hPGCLCs (WT, left; KO, right) for AGVT expression. (**F**) The number of total cells, or AG$^+$ or AG$^-$ cells in hPGCLC aggregates at day5. Statistically significant differences between WT (black circles) and KO (white circles) were identified with a two-tailed t-test. Means ± standard deviation are shown. n.s., not significant.
(PDF)

**S1 Table. The list of sgRNAs targeting 422 coding-genes and 50 non-targeting control sgRNAs.**
(XLSX)

**S2 Table. Screening metrics for all samples in PGCLC regulator screen.**
(XLSX)

**S3 Table. Screening results by comparing PGCLC(+) vs. [PGCLC(-) and iMeLC].**
(XLSX)

**S4 Table. Differentially expressed genes between PGCLC_WT and PGCLC_KO by DESeq2 (FDR-adjusted p-value ≤ 0.05).**
(XLSX)

**S5 Table. Gene Ontology enrichment analysis of DEGs downregulated in *TCL1A* KO PGCLCs (KO down) (p.value ≤ 0.05).**
(XLSX)

**S6 Table. Gene Ontology enrichment analysis of DEGs upregulated in *TCL1A* KO PGCLCs (KO up) (p.value ≤ 0.05).**
(XLSX)

**S7 Table. Primers used in this study.**
(XLSX)

## Acknowledgments

We thank L. King for carefully reviewing the manuscript and providing insightful comments. We appreciate V. Nallamala for scRNA-seq analysis. We thank members of Sasaki and Blanco laboratory for discussion of this study.

## Author Contributions

**Conceptualization:** Young Sun Hwang, Kotaro Sasaki.

**Data curation:** Young Sun Hwang.

**Formal analysis:** Yasunari Seita.

**Funding acquisition:** Kotaro Sasaki.

**Investigation:** Young Sun Hwang, M. Andrés Blanco, Kotaro Sasaki.

**Methodology:** M. Andrés Blanco.

**Project administration:** Kotaro Sasaki.

**Resources:** Young Sun Hwang.

**Supervision:** M. Andrés Blanco, Kotaro Sasaki.

**Validation:** Young Sun Hwang, Kotaro Sasaki.

**Visualization:** Young Sun Hwang, M. Andrés Blanco.

**Writing – original draft:** Young Sun Hwang, Kotaro Sasaki.

**Writing – review & editing:** M. Andrés Blanco, Kotaro Sasaki.

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
