## [Decision Letter · Decision Letter 0]

24 Oct 2023

Dear Dr Sasaki,

Thank you very much for submitting your Research Article entitled 'CRISPR loss of function screening to identify genes involved in human primordial germ cell-like cell development' to PLOS Genetics.

The manuscript was evaluated by the editorial board and a single reviewer in the context of the previous comments received at Review Commons. As you will see, we are interested in moving forward but ask that you address some remaining concerns in a revised manuscript.

We therefore ask you to modify the manuscript according to the review recommendations. Your revisions should address the specific points made by each reviewer.

In addition we ask that you provide a detailed list of your responses to the review comments and a description of the changes you have made in the manuscript.

Yours sincerely,

Gregory S. Barsh

Editor-in-Chief

PLOS Genetics

Gregory Copenhaver

Editor-in-Chief

PLOS Genetics

Reviewer's Responses to Questions

**Comments to the Authors:**

Reviewer #1: In this manuscript, Hwang et al perform a CRISPR loss of function screen using a human PGCLC induction system. Amongst the candidate genes which the authors identified, they focus on TCL1A and METTLE7. Regarding TCL1A, the authors find its function in hPGCLC survival or proliferation but not specification, and the possible role in translational regulation during hPGCLC development. Furthermore, the authors demonstrate that TCL1A acts as a regulator of AKT signaling, which partly explains the impairment of proliferation in TCL1A KO hPGCLCs. The application of CRISPR screen using the PGCLC induction system is reported in mice but novel in humans, which is a technical advancement of this study, whereas the number of genes used in this study is limited. In response to comments from the other reviewers, the manuscript seems to be improved in terms of reproducibility and statistics, but I think there remain some concerns to address, as follows.

1. TCL1A expression and function in non-hPGCLC cells in embryoid body

As the other reviewers pointed out, loss of TCL1A likely impacts non-hPGCLC cells in embryoid bodies. In the revision, the authors have added scRNA-seq data to claim the specific expression in germ cells during male germ cell development, but it does not address whether non-hPGCLC cells in the embryoid bodies express TCL1A or not. Thus, there remains a concern that TCL1A might not be specific to hPGCLCs but rather a downstream target of cytokines used in the induction. The authors should confirm the expression by using their own or public scRNA-seq dataset, or immunostaining. If the expression is not specific in hPGCLCs, they should discuss it as a general role in early embryo development.

2. Lack of some key regulators for PGC development in the candidate genes

The authors claim that their CRISPR screening is working because hit candidates contain SOX17 and TFAP2C, both are critical transcriptional regulators for hPGCLC specification and development. So far, the author’s group and the others identified some other key regulators such as EOMES and PRDM1, which are included in a list of CRISPR screen targets, but they seem to be lower scores from the result of the screen. The authors should explain how it happened and discuss what are the technical limitations of their system.

3. Gene expression pattern of TCL1A knockout hPGCLCs

In the comparative analysis of hPGCLCs transcriptome between wildtype control and TCL1A knockout, the authors only show PCA and DEG analysis. To clarify TCL1A knockout does not affect key germ cell regulators, it would be better to show a representative gene expression dataset based on categories such as pluripotency, germ cells, and epigenetics. It is curious if the loss of TCL1A affects genes upregulated after specification such as KLF4 and TFCP2L1.

4. In Figure 4A and B, SOX17 alone is not sufficient to detect hPGCLCs.

Since SOX17 is also expressed in definitive endoderm, the use of SOX17 alone as a marker for hPGCLCs is not appropriate. A loss of TCL1A may also affect the proportion of cell types in embryoid bodies.

**Have all data underlying the figures and results presented in the manuscript been provided?**

Reviewer #1: Yes

PLOS authors have the option to publish the peer review history of their article (what does this mean?). If published, this will include your full peer review and any attached files.

Reviewer #1: No

---

## [Editor Report · Decision Letter 1]

22 Nov 2023

Dear Dr Sasaki,

We are pleased to inform you that your manuscript entitled "CRISPR loss of function screening to identify genes involved in human primordial germ cell-like cells development" has been editorially accepted for publication in PLOS Genetics. Congratulations!

Yours sincerely,

Gregory S. Barsh

Editor-in-Chief

PLOS Genetics

Gregory Copenhaver

Editor-in-Chief

PLOS Genetics

Comments from the reviewers (if applicable):

**Data Deposition**

http://datadryad.org/submit?journalID=pgenetics&manu=PGENETICS-D-23-01001R1

**Press Queries**

---

## [Editor Report · Acceptance letter]

7 Dec 2023

PGENETICS-D-23-01001R1 

CRISPR loss of function screening to identify genes involved in human primordial germ cell-like cell development 

Dear Dr Sasaki, 

We are pleased to inform you that your manuscript entitled "CRISPR loss of function screening to identify genes involved in human primordial germ cell-like cell development" has been formally accepted for publication in PLOS Genetics! Your manuscript is now with our production department and you will be notified of the publication date in due course.

With kind regards,

Judit Kozma

PLOS Genetics

On behalf of:
